# Spectral methods for neural characterization using generalized quadratic models

**Il Memming Park**[*123], **Evan Archer**[*13], **Nicholas Priebe**[14], **& Jonathan W. Pillow**[123]
1. Center for Perceptual Systems, 2. Dept. of Psychology,
3. Division of Statistics & Scientific Computation, 4. Section of Neurobiology,
The University of Texas at Austin
{memming@austin., earcher@, nicholas@, pillow@mail.} utexas.edu

## Abstract

We describe a set of fast, tractable methods for characterizing neural responses to high-dimensional sensory stimuli using a model we refer to as the generalized quadratic model (GQM). The GQM consists of a low-rank quadratic function followed by a point nonlinearity and exponential-family noise. The quadratic function characterizes the neuron's stimulus selectivity in terms of a set linear receptive fields followed by a quadratic combination rule, and the invertible nonlinearity maps this output to the desired response range. Special cases of the GQM include the 2nd-order Volterra model [1, 2] and the elliptical Linear-Nonlinear-Poisson model [3]. Here we show that for "canonical form" GQMs, spectral decomposition of the first two response-weighted moments yields approximate maximum-likelihood estimators via a quantity called the *expected log-likelihood*. The resulting theory generalizes moment-based estimators such as the spike-triggered covariance, and, in the Gaussian noise case, provides closed-form estimators under a large class of non-Gaussian stimulus distributions. We show that these estimators are fast and provide highly accurate estimates with far lower computational cost than full maximum likelihood. Moreover, the GQM provides a natural framework for combining multi-dimensional stimulus sensitivity and spike-history dependencies within a single model. We show applications to both analog and spiking data using intracellular recordings of V1 membrane potential and extracellular recordings of retinal spike trains.

## 1 Introduction

Although sensory stimuli are high-dimensional, sensory neurons are typically sensitive to only a small number of stimulus features. Linear dimensionality-reduction methods seek to identify these features in terms of a subspace spanned by a small number of spatiotemporal filters. These filters, which describe how the stimulus is integrated over space and time, can be considered the first stage in a "cascade" model of neural responses. In the well-known *linear-nonlinear-Poisson* (LNP) cascade model, filter outputs are combined via a nonlinear function to produce an instantaneous spike rate, which generates spikes via an inhomogeneous Poisson process [4, 5].

The most popular methods for dimensionality reduction with spike train data involve the first two moments of the spike-triggered stimulus distribution: (1) the spike-triggered average (STA) [7–9]; and (2) major and minor eigenvectors of spike-triggered covariance (STC) matrix [10, 11][1]. STC analysis can be described as a *spectral method* because the estimate is obtained by eigenvector

---

[*] These authors contributed equally.

[1]Related moment-based estimators have also appeared in the statistics literature under the names "inverse regression" and "sufficient dimensionality reduction", although the connection to STA and STC analysis does not appear to have been noted previously [12, 13].

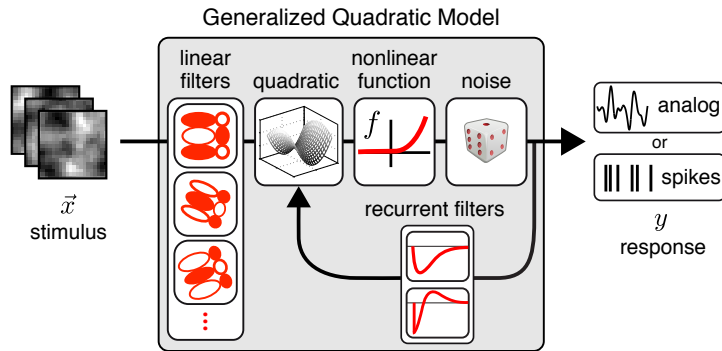

Figure 1: Schematic of generalized quadratic model (GQM) for analog or spike train data.

decomposition of an appropriately defined matrix. Compared to likelihood-based methods, spectral methods are generally computationally efficient and devoid of (non-global) local optima.

Recently, Park and Pillow [3] described a connection between STA/STC analysis and maximum likelihood estimators based on a quantity called the *expected log-likelihood* (EL). The EL results from replacing the nonlinear term in the log-likelihood and with its expectation over the stimulus distribution. When the stimulus is Gaussian, the EL depends only on moments (mean spike rate, STA, STC, and stimulus mean and covariance) and leads to a closed-form spectral estimate for LNP filters, which has STC analysis as a special case. More recently, Ramirez and Paninski derived EL-based estimators for the linear Gaussian model and proposed fast EL-based inference methods for generalized linear models (GLMs) [14].

Here, we show that the EL framework can be extended to a more general class that we refer to as the *generalized quadratic model* (GQM). The GQM represents a straightforward extension of the generalized linear model GLM [15, 16] wherein the linear predictor is replaced by a quadratic function (Fig. 1). For Gaussian and Poisson GQMs, we derive computationally efficient EL-based estimators that apply to a variety of non-Gaussian stimulus distributions; this substantially extends previous work on the conditions of validity for moment-based estimators [7, 17–19]. In the Gaussian case, the EL-based estimator has a closed form solution that relies only on the first two response-weighted moments and the first four stimulus moments. In the Poisson case, GQMs provide a natural synthesis of models that have multiple filters (i.e., where the response depends on multiple projections of the stimulus) and dependencies on spike history. We show that spectral estimates of a low-dimensional feature space are nearly as accurate as maximum likelihood estimates (for GQMs without spike-history), and demonstrate the applicability of GQMs for both analog and spiking data.

## 2 Generalized Quadratic Models

We begin by briefly reviewing of the class of models known as GLMs, which includes the single-filter LNP model, and the Wiener model from the systems identification literature. A GLM has three basic components: a linear stimulus filter, an invertible nonlinearity (or "inverse link" function), and an exponential-family noise model. The GLM describes the conditional response $y$ to a vector stimulus $\mathbf{x}$ as:

$$y|\mathbf{x} \sim \mathbf{P}(f(\mathbf{w}^\top \mathbf{x})), \tag{1}$$

where $\mathbf{w}$ is the filter, $f$ is the nonlinearity, and $\mathbf{P}(\lambda)$ denotes a noise distribution function with mean $\lambda$. From the standpoint of dimensionality reduction, the GLM makes the strong modeling assumption that response $y$ depends upon $\mathbf{x}$ only via its one-dimensional projection onto $\mathbf{w}$.

At the other end of the modeling spectrum sits the very general "multiple filter" linear-nonlinear (LN) cascade model, which posits that the response depends on a $p$-dimensional projection of the stimulus, represented by a bank of filters $\{\mathbf{w}_i\}_{i=1}^p$, and combined via some arbitrary multi-dimensional function $f : \mathbb{R}^p \to \mathbb{R}$:

$$y|\mathbf{x} \sim \mathbf{P}(f(\mathbf{w}_1^\top \mathbf{x}, \dots, \mathbf{w}_p^\top \mathbf{x})). \tag{2}$$

Spike-triggered covariance analysis and related methods provide low-cost estimates of the filters $\{\mathbf{w}_i\}$ under Poisson or Bernoulli noise models, but only under restrictive conditions on the stimulus

distribution (e.g., elliptical symmetry) and some weak conditions on $f$ [17, 19]. Semi-parametric estimators like "maximally informative dimensions" (MID) eliminate these restrictions [20], but do not practically scale beyond two or three filters without additional modeling assumptions [21].

The generalized quadratic model (GQM) provides a tractable middle ground between the GLM and general multi-filter LN models. The GQM allows for multi-dimensional stimulus dependence, yet restricts the nonlinearity to be a transformed quadratic function [22–25]. The GQM can be written:

$$y|\mathbf{x} \sim \mathbf{P}(f(Q(\mathbf{x}))), \tag{3}$$

where $Q(\mathbf{x}) = \mathbf{x}^\top C \mathbf{x} + \mathbf{b}^\top \mathbf{x} + a$ denotes a quadratic function of $\mathbf{x}$, governed by a (possibly low-rank) symmetric matrix $C$, a vector $\mathbf{b}$, and a scalar $a$. Note that the GQM may be regarded as a GLM in the space of quadratically transformed stimuli [6], although this approach does not allow $Q(\mathbf{x})$ to be parametrized directly in terms of a projection onto a small number of linear filters.

In the following, we show that the elliptical-LNP model [3] is a GQM with Poisson noise, and make a detailed study of canonical GQMs with Gaussian noise. We show that the maximum-EL estimates for $C$, $\mathbf{b}$, and $a$ have similar forms for both Gaussian and Poisson GQMs, and that the eigenspectrum of $C$ provides accurate estimates of a neuron's low-dimensional feature space. Finally, we show that the GQM provides a natural framework for combining multi-dimensional stimulus sensitivity with dependencies on spike train history or other response covariates.

## 3 Estimation with expected log-likelihoods

The expected log-likelihood is a quantity that approximates log-likelihood but can be computed very efficiently using moments. It exists for any GQM or GLM with "canonical" nonlinearity (or link function). The canonical nonlinearity for an exponential-family noise distribution has the special property that it allows the log-likelihood to be written as the sum of two terms: a term that depends linearly on the responses $\{y_i\}$, and a second (nonlinear) term that depends only on the stimuli $\{\mathbf{x}_i\}$ and parameters $\theta$. The expected log-likelihood (EL) results from replacing the nonlinear term with its expectation over the stimulus distribution $P(\mathbf{x})$, which in neurophysiology settings is often known *a priori* to the experimenter. Maximizing the EL results in *maximum expected log-likelihood* (MEL) estimators that have very low computational cost while achieving nearly the accuracy of full maximum likelihood (ML) estimators. Spectral decompositions derived from the EL provide estimators that generalize STA/STC analysis. In the following, we derive MEL estimators for three special cases—two for the Gaussian noise model, and one for the Poisson noise model.

### 3.1 Gaussian GQMs

Gaussian noise provides a natural model for analog neural response variables like membrane potential or fluorescence. The canonical nonlinearity for Gaussian noise is the identity function, $f(x) = x$. The the canonical-form Gaussian GQM can therefore be written: $y|\mathbf{x} \sim \mathcal{N}(Q(\mathbf{x}), \sigma^2)$. Given a dataset $\{\mathbf{x}_i, y_i\}_{i=1}^N$, the log-likelihood per sample is:

$$\mathcal{L} = -\frac{1}{2\sigma^2}\frac{1}{N}\sum_i \left(Q(\mathbf{x}_i) - y_i\right)^2 = -\frac{1}{2\sigma^2}\frac{1}{N}\sum_i \left(-2Q(\mathbf{x}_i)y_i + Q(\mathbf{x}_i)^2\right) + const$$

$$= -\frac{1}{2\sigma^2}\left(-2\left(\mathrm{Tr}(C\Lambda) + \mu^\top \mathbf{b} + a\bar{y}\right) + \frac{1}{N}\sum_i Q(\mathbf{x}_i)^2\right) + const, \tag{4}$$

where $\sigma^2$ is the noise variance, $const$ is a parameter-independent constant, $\bar{y} = \frac{1}{N}\sum_i y_i$ is the mean response, and $\mu$ and $\Lambda$ denote cross-correlation statistics that we will refer to (in a slight abuse of terminology) as the *response triggered average* and *response-triggered covariance*:

$$\mu = \frac{1}{N}\sum_{i=1}^N y_i\mathbf{x}_i \text{ ("RTA")} \qquad \Lambda = \frac{1}{N}\sum_{i=1}^N y_i\mathbf{x}_i\mathbf{x}_i^\top \text{ ("RTC").}^2 \tag{5}$$

The expected log-likelihood results from replacing the troublesome nonlinear term $\frac{1}{N}\sum_i Q(\mathbf{x}_i)^2$ by its expectation over the stimulus distribution. This is justified by the law of large numbers, which

asserts that $\frac{1}{N}\sum_i Q(\mathbf{x}_i)^2$ converges to $\mathbb{E}_{P(\mathbf{x})}[Q(\mathbf{x})^2]$ asymptotically. Leaving off the *const* term, this leads to the per-sample *expected log-likelihood* [3, 14], which is defined:

$$\tilde{\mathcal{L}} = -\frac{1}{2\sigma^2}\left(-2\left(\text{Tr}(C\Lambda) + \mu^\top \mathbf{b} + a\bar{y}\right) + \mathbb{E}[Q(\mathbf{x})^2]\right). \tag{6}$$

**Gaussian stimuli**
If the stimuli are drawn from a Gaussian distribution, $\mathbf{x} \sim \mathcal{N}(0, \Sigma)$, then we have (from [26]):

$$\mathbb{E}[Q(\mathbf{x})^2] = 2\,\text{Tr}\left\{(C\Sigma)^2\right\} + \text{Tr}(\mathbf{b}^T\Sigma\mathbf{b}) + (\text{Tr}(C\Sigma) + a)^2. \tag{7}$$

The EL is concave in the parameters $a, \mathbf{b}, C$, so we can obtain the MEL estimates by finding the stationary point:

$$\frac{\partial}{\partial a}\tilde{\mathcal{L}} = -\frac{1}{2\sigma^2}\left(-2\bar{y} + 2\left(\text{Tr}(C\Sigma) + a\right)\right) = 0 \quad \Longrightarrow \quad a_{\text{mel}} = \bar{y} - \text{Tr}(C_{\text{mel}}\Sigma)) \tag{8}$$

$$\frac{\partial}{\partial \mathbf{b}}\tilde{\mathcal{L}} = -\frac{1}{2\sigma^2}\left(-2\mu + 2\Sigma\mathbf{b}\right) = 0 \quad \Longrightarrow \quad \mathbf{b}_{\text{mel}} = \Sigma^{-1}\mu \tag{9}$$

$$\frac{\partial}{\partial C}\tilde{\mathcal{L}} = -\frac{1}{2\sigma^2}\left(-2\Lambda + \left(4\Sigma C\Sigma + 2\bar{y}\Sigma\right)\right) = 0 \quad \Longrightarrow \quad C_{\text{mel}} = \frac{1}{2}\left(\Sigma^{-1}\Lambda\Sigma^{-1} - \bar{y}\Sigma^{-1}\right) \tag{10}$$

Note that this coincides with the moment-based estimate for the 2nd-order Volterra model [2].

**Axis-symmetric stimuli**
More generally, we can derive the MEL estimator for stimuli with arbitrary axis-symmetric distributions with finite 4th-order moments. Axis-symmetric distributions exhibit invariance under reflections around each axis, that is, $P(x_1, \ldots, x_d) = P(\rho_1 x_1, \ldots, \rho_d x_d)$ for any $\rho_i \in \{-1, 1\}$. The class of axis-symmetric distributions subsumes both radially symmetric and independent product distributions. However, axis symmetry is a strictly weaker condition; significantly, marginals need not be identically distributed.

To simplify derivation of the MEL estimator for axis-symmetric stimuli, we take the derivative of $Q(\mathbf{x})$ with respect to $(a, \mathbf{b}, C)$ before taking the expectation. Derivatives with respect to model parameters are given by $\frac{\partial\mathbb{E}[Q(\mathbf{x})^2]}{\partial\theta_i} = \mathbb{E}\left[2Q(\mathbf{x})\frac{\partial Q(\mathbf{x})}{\partial\theta_i}\right]$. For each $\theta_i$, we solve the equation,

$$\frac{\partial\tilde{\mathcal{L}}}{\partial\theta_i} = -2\frac{\partial\left(\text{Tr}(C\Lambda) + \mu^\top\mathbf{b} + a\bar{y}\right)}{\partial\theta_i} + 2\mathbb{E}\left[Q(\mathbf{x})\frac{\partial Q(\mathbf{x})}{\partial\theta_i}\right] = 0.$$

From derivatives w.r.t. $a$, $\mathbf{b}$, and $C$, respectively, we obtain conditions for the MEL estimates:

$$\bar{y} = \mathbb{E}\left[Q(\mathbf{x})\right] = a + \mathbf{b}^\top\mathbb{E}[\mathbf{x}] + \text{Tr}(C\mathbb{E}[\mathbf{x}\mathbf{x}^\top])$$

$$\mu = \mathbb{E}\left[Q(\mathbf{x})\mathbf{x}\right] = a\mathbb{E}[\mathbf{x}] + \mathbf{b}^\top\mathbb{E}[\mathbf{x}\mathbf{x}^\top] + \sum_{i,j}C_{ij}\mathbb{E}[\mathbf{x}_i\mathbf{x}_j\mathbf{x}]$$

$$\Lambda = \mathbb{E}\left[Q(\mathbf{x})\mathbf{x}\mathbf{x}^\top\right] = a\mathbb{E}[\mathbf{x}\mathbf{x}^\top] + \sum_i b_i\mathbb{E}[\mathbf{x}_i\mathbf{x}\mathbf{x}^\top] + \sum_{i,j}C_{ij}\mathbb{E}[\mathbf{x}_i\mathbf{x}_j\mathbf{x}\mathbf{x}^\top]$$

where the subindices within the sums are for components. Due to axis symmetry, $\mathbb{E}[\mathbf{x}]$, $\mathbb{E}[x_i x_j x_k]$ and $\mathbb{E}[x_i x_j^3]$ are all zero for distinct indices. Thus, the MEL estimates for $a$ and $\mathbf{b}$ are identical to the Gaussian case given above. If we further assume that the stimulus is whitened so that $\mathbb{E}[\mathbf{x}\mathbf{x}^\top] = \mathbf{I}$, sufficient stimulus statistics are the 4th order even moments, which we represent with the matrix $M_{ij} = \mathbb{E}\left[x_i^2 x_j^2\right]$.

In general, when the marginals are not identical but the joint distribution is axis-symmetric,

$$\sum_{ij}C_{ij}\mathbb{E}[x_i x_j \mathbf{x}\mathbf{x}^\top] = \sum_i C_{ii}\,\text{diag}(x_i^2 x_1^2, \cdots, x_i^2 x_d^2) + \sum_{i \neq j}C_{ij}M_{ij}\mathbf{e}_i\mathbf{e}_j^\top \tag{11}$$

$$= \text{diag}(\mathbf{1}^\top(\mathbb{I} \circ C)M) + C \circ M \circ (\mathbf{1}\mathbf{1}^\top - \mathbb{I}).$$

where $\mathbf{1}$ is a vector of 1's, $\mathbf{e}_i$ is the standard basis, and $\circ$ denotes the Hadamard product. We can solve these sets of linear equations for the diagonal terms and off-diagonal terms separately obtaining,

$$[C_{\text{mel}}]_{ij} = \begin{cases} \frac{\Lambda_{ij}}{2M_{ij}}, & i \neq j \\ \Omega(M - \mathbf{1}\mathbf{1}^\top)^{-1}, & i = j \end{cases} \tag{12}$$

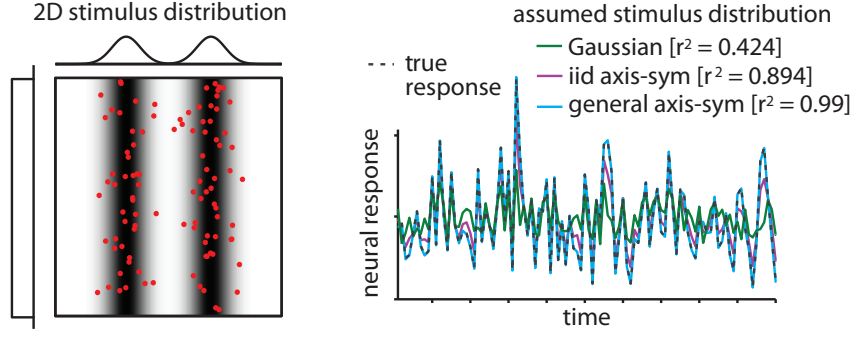

Figure 2: Maximum expected log-likelihood (MEL) estimators for a Gaussian GQM under different assumptions about the stimulus distribution. **(left)** Axis-symmetric stimulus distribution in 2D. The horizontal axis is a (symmetric) mixture of Gaussian, and the vertical axis is a uniform distribution. Red dots indicate samples from the distribution. **(right)** Response prediction based on various $\hat{C}$ estimated using eq. 10, eq. 14, and eq. 12. Performance is evaluated on a cross-validation test set with no noise for each $C$, and we see a huge loss in performance as a result of incorrect assumption about the stimulus distribution.

where $\Omega = \mathrm{diag}(\mathbf{1}^\top (\mathbb{I} \circ \Lambda) - \bar{y}\mathbf{1}^\top)$.

For the special case when the marginal distributions are identical, we note that

$$\mathbb{E}[\mathbf{x}^\top C \mathbf{x}(\mathbf{x}\mathbf{x}^\top)] = \mu_{22} \mathrm{Tr}(C)\mathbb{I} + (\mu_4 - \mu_{22})C \circ \mathbb{I} + 2\mu_{22}C \circ (\mathbf{1}\mathbf{1}^\top - \mathbb{I}) \tag{13}$$

where $\mu_{22} = \mathbb{E}[\mathbf{x}_1^2 \mathbf{x}_2^2] = M_{1,2}$ and $\mu_4 = \mathbb{E}[\mathbf{x}_1^4] = M_{1,1}$. This gives the simplified formula (also given in [27]):

$$[C_{\mathrm{mel}}]_{ij} = \begin{cases} \frac{\Lambda_{ij}}{2\mu_{22}}, & i \neq j \\ \frac{\Lambda_{ii} - \bar{y}}{\mu_4 - \mu_{22}}, & i = j \end{cases} \tag{14}$$

When the stimulus is not Gaussian or the marginals not identical, the estimates obtained from (eq. 10) and (eq. 14) are not consistent. In this case, the general axis-symmetric estimate (eq. 12) gives much better performance, as we illustrate with a simulated example in Fig. 2.

### 3.2 Poisson GQM

Poisson noise provides a natural model for discrete events like spike counts, and extends easily to point process models for spike trains. The canonical nonlinearity for Poisson noise is exponential, $f(x) = \exp(x)$, so the canonical-form Poisson GQM is: $y|\mathbf{x} \sim \mathrm{Poiss}(\exp(Q(\mathbf{x})))$. Ignoring irrelevant constants, the log-likelihood per sample is

$$\mathcal{L} = \frac{1}{N}\sum_i y_i \log(\exp(Q(\mathbf{x}_i))) - \frac{1}{N}\sum_i \exp(Q(\mathbf{x}_i))$$
$$= \mathrm{Tr}(C\Lambda) + \mu^\top \mathbf{b} + a\bar{y} - \frac{1}{N}\sum_i \exp(Q(\mathbf{x}_i)), \tag{15}$$

where $\bar{y}$, $\mu$ and $\Lambda$ denote mean response, STA, and STC, as given above (eq. 5). We obtain the EL for a Poisson GQM by replacing the term $\frac{1}{N}\sum \exp(Q(\mathbf{x}_i))$ by its expectation with respect to $P(\mathbf{x})$. Under a zero-mean Gaussian stimulus distribution with covariance $\Sigma$, the closed-form MEL estimates are (from [3]):

$$\mathbf{b}_{\mathrm{mel}} = \left(\Lambda + \frac{1}{\bar{y}}2\mu\mu^\top\right)^{-1}\mu, \qquad C_{\mathrm{mel}} = \frac{1}{2}\left(\Sigma^{-1} - \bar{y}\left(\Lambda + \frac{1}{\bar{y}}2\mu\mu^\top\right)^{-1}\right), \tag{16}$$

where we assume that $\Lambda + \frac{1}{\bar{y}}2\mu\mu^\top$ is invertible. Note that the MEL estimator combines information from $\mu$ and $\Lambda$, unlike standard STA and STC-based estimates, which maximize EL only when either $\mathbf{b}$ or $C$ is zero (respectively). Park and Pillow 2011 used Poisson EL in conjunction with a log-prior to obtain approximate Bayesian estimates, an approach referred to as *Bayesian STC* [3].

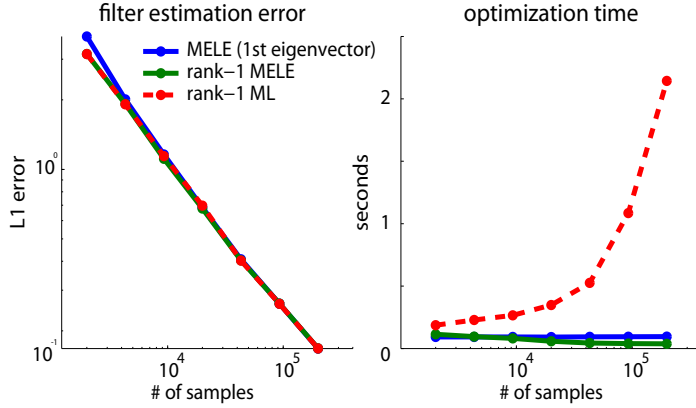

Figure 3: Rank-1 quadratic filter reconstruction performance. Both rank-1 models were optimized using conjugate gradient descent. (Left) $l_1$ distance from the ground truth filter. (Right) Computation time for the optimization.

**Mixture-of-Gaussians stimuli**

Results for Gaussian stimuli extend naturally to mixtures of Gaussians, which can be used to approximate arbitrary stimulus distributions. The EL for mixture-of-Gaussian stimuli can be computed simply via the linearity of expectation. For stimuli drawn from a mixture $\sum_i \alpha_j \mathcal{N}(\mu_j, \Sigma_j)$ with mixing weights $\sum_j \alpha_j = 1$, the EL is

$$\tilde{\mathcal{L}} = \text{Tr}(C\Lambda) + \mu^\top \mathbf{b} + a\bar{y} - \sum_i \alpha_j \mathbb{E}_{\mathcal{N}(\mu_j, \Sigma_j)}[e^{Q(\mathbf{x})}], \tag{17}$$

where the Gaussian expectation terms are given by

$$\mathbb{E}_{\mathcal{N}(\mu_j, \Sigma_j)}[e^{Q(\mathbf{x})}] = \frac{1}{|I - 2C\Sigma_j|^{\frac{1}{2}}} e^{\left(a + \mu_j^\top C \mu_j + \mathbf{b}^\top \mu_j + \frac{1}{2}(\mathbf{b} + 2C\mu_j)^\top (\Sigma_j^{-1} - 2C)^{-1}(\mathbf{b} + 2C\mu_j)\right)}. \tag{18}$$

Although the MEL estimator does not have a closed analytic form in this case, the EL can be efficiently optimized numerically, as it still depends on the responses only via the spike-triggered moments $\bar{y}$, $\mu$ and $\Lambda$, and on the stimuli only via the mean, covariance, and mixing weight of each Gaussian.

# 4 Spectral estimation for low-dimensional models

## 4.1 Low-rank parameterization

We have so far focused upon MEL estimators for the parameters $a$, $\mathbf{b}$, and $C$. These results have a natural mapping to dimensionality reduction methods. Under the GQM, a low-dimensional stimulus dependence is equivalent to having a low-rank $C$. If $C = BB^\top$ for some $d \times p$ matrix $B$, we have a $p$-filter model (or $p+1$ filter model if the linear term $\mathbf{b}$ is not spanned by the columns of $B$). We can obtain spectral estimates of a low-dimensional GQM by performing an eigenvector decomposition of $C_{\text{mel}}$ and selecting the eigenvectors corresponding to the largest $p$ eigenvalues. The eigenvectors of $C_{\text{mel}}$ also make natural initializers for maximization of the full GQM likelihood.

In Fig. 3, we show the results of three different methods for recovering a simulated rank-1 GQM with Poisson noise: (1) the largest eigenvector of $C_{\text{mel}}$, (2) numerically maximizing the expected log-likelihood for a rank-1 GQM (i.e., with $C$ parametrized as a rank-1 matrix), and (3) maximizing the (full) likelihood for a rank-1 GQM. Although the difference in performance between expected and full GQM log-likelihood is negligible, there is a drastic difference in optimization time complexity between the full and expected log-likelihood. The expected log-likelihood only requires computation of the sufficient statistics, while the full ML estimate requires a full pass through the dataset for each evaluation of the log-likelihood. Thus, the expected log-likelihood offers a fast yet accurate estimate for $C$. In the following section we show that, asymptotically, the eigenvectors of $C_{\text{mel}}$ span the "correct" (in an appropriate sense) low-dimensional subspace.

## 4.2 Consistency of subspace estimates

If the conditional probability $y|\mathbf{x} = y|\boldsymbol{\beta}^\top \mathbf{x}$ for a matrix $\boldsymbol{\beta}$, the neural feature space is spanned by the columns of $\boldsymbol{\beta}$. As a generalization of STC, we introduce moment-based dimensionality reduction

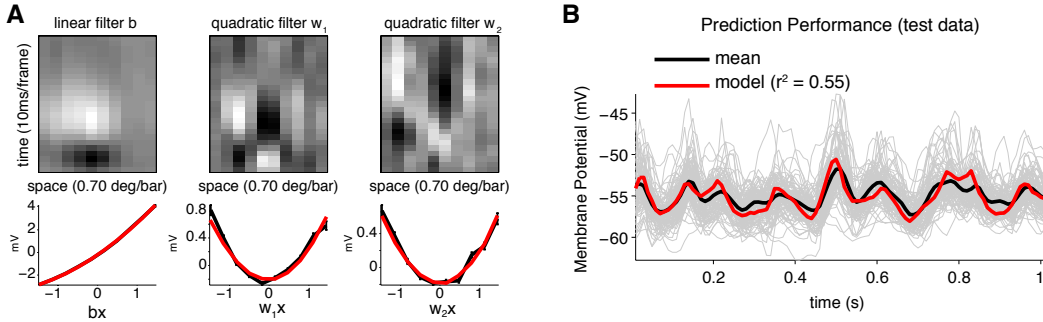

Figure 4: GQM fit and prediction for intracellular recording in cat V1 with a trinary noise stimulus.
**(A)** On top, estimated linear ($b$) and quadratic ($w_1$ and $w_2$) filters for the GQM, lagged by 20ms. On bottom, the empirical marginal nonlinearities along each dimension (black) and model prediction (red). **(B)** Cross-validated model prediction (red) and $n = 94$ recordings with repeats of identical stimulus (light grey) along with their mean (black). Reported performance metric ($r^2 = 0.55$) is for prediction of the mean response.

techniques that recover (portions of) $\boldsymbol{\beta}$ and show the relationship of these techniques to the MEL estimators of GQM.

We propose to use $\Sigma^{-\frac{1}{2}}\mu$ and eigenvectors of $\Sigma^{-\frac{1}{2}}\Lambda\Sigma^{-\frac{1}{2}}$ (whose eigenvalues are significantly smaller or larger than 1) as the feature space basis. When the response is binary, this coincides with the traditional STA/STC analysis, which is provably consistent only in the case of stimuli drawn from a spherically symmetric (for STA) or independent Gaussian distribution (for STC) [5].

Below, we argue that this procedure can identify the subspace when $y$ has mean $f(\boldsymbol{\beta}^\top \mathbf{x})$ with finite variance, $f$ is some function, and the stimulus distribution is zero-mean with white covariance, i.e., $\mathbb{E}[\mathbf{x}] = 0$ and $\mathbb{E}[\mathbf{x}\mathbf{x}^T] = I$.

First, note that by the law of large numbers, $\Lambda \to \mathbb{E}[y\,\mathbf{x}\mathbf{x}^T] = \mathbb{E}\left[y\mathbb{E}[\mathbf{x}\mathbf{x}^T|y]\right]$. Let $\Psi = \boldsymbol{\beta}\boldsymbol{\beta}^T$ be a projection operator to the feature space, and $\Psi_\perp = I - \Psi$ be the perpendicular space. We follow the discussion in [12, 13] regarding the related "sliced regression" literature. Recalling that $\mathbb{E}[X] = 0$, we can exploit the independence of $\Psi_\perp \mathbf{x}$ and $y$ to find,

$$\mathbb{E}\left[\mathbf{x}\mathbf{x}^\top|y = \xi\right] = \mathbb{E}\left[(\Psi + \Psi_\perp)\mathbf{x}\mathbf{x}^\top(\Psi + \Psi_\perp)|y = \xi\right]$$
$$= \Psi\mathbb{E}\left[\mathbf{x}\mathbf{x}^\top|y = \xi\right]\Psi + \Psi_\perp\mathbb{E}\left[\mathbf{x}\mathbf{x}^\top\right]\Psi_\perp = \Psi\mathbb{E}\left[\mathbf{x}\mathbf{x}^\top|y = \xi\right]\Psi + \Psi_\perp$$

thus, $\mathbb{E}\left[y\mathbf{x}\mathbf{x}^\top\right] = \Psi\mathbb{E}\left[y\mathbf{x}\mathbf{x}^\top\right]\Psi + \mathbb{E}[y]\Psi_\perp$ and therefore the eigenvectors of $\mathbb{E}\left[y\mathbf{x}\mathbf{x}^\top\right]$ whose eigenvalues significantly differ from $\mathbb{E}[y]$ span a subspace of the range of $\Psi$. Effective estimation of the subspace depends critically on both the stimulus distribution and the form of $f$. Under the GQM, the eigenvectors of $\mathbb{E}\left[y\mathbf{x}\mathbf{x}^\top\right]$ are closely related to the expected log-likelihood estimators we derived earlier. Indeed, those eigenvectors of eq. 10, eq. 12 and eq. 16 whose associated eigenvalues differ significantly from zero span precisely the same space.

# 5 Results

## 5.1 Intracellular membrane potential

We fit a Gaussian GQM to intracellular recordings of membrane potential from a neuron in cat V1, using a 2D spatiotemporal "flickering bars" stimulus aligned with the cell's preferred orientation (Fig. 4). The recorded time-series is a continuous signal, so the Gaussian GQM provides an appropriate noise model. The recorded voltage was median-filtered (to remove spikes) and down-sampled to a 10 ms sample rate. We fit the GQM to a 21.6 minute recording of responses to non-repeating trinary noise stimulus . We validated the model using responses to 94 repeats of a 1 second frozen noise stimulus. Panel **(B)** of Fig. 4 illustrates the GQM prediction on cross-validation data.

Although the cell was classified as "simple", meaning that its response is predominately linear, the GQM fit reveals two quadratic filters that also influence the membrane potential response. The GQM captures a substantial percentage of the variance in the mean response, systematically outperforming the GLM in terms of $r^2$ (GQM:55% vs. GLM:50%).

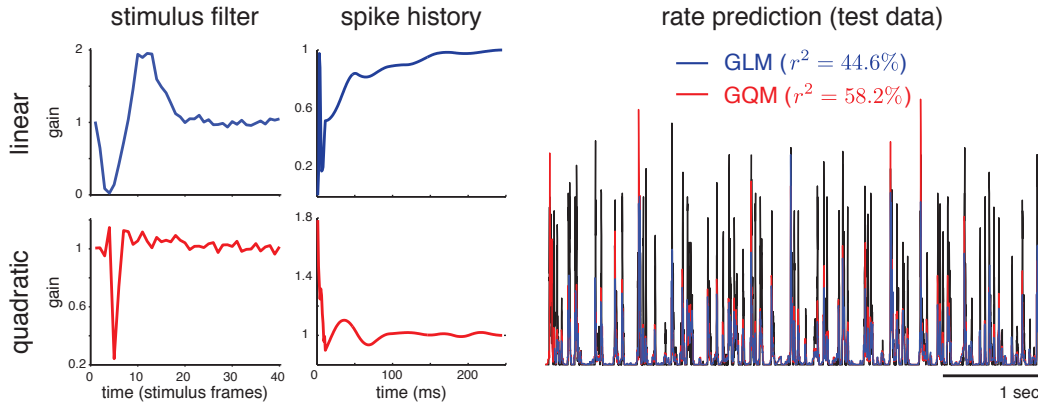

Figure 5: **(left)** GLM and GQM filters fit to spike responses of a retinal ganglion cell stimulated with a 120 Hz binary full field noise stimulus [28]. The GLM has only linear stimulus and spike history filters (top left) while the GQM contains all four filters. Each plot shows the exponentiated filter, so the ordinate has units of gain, and filters interact multiplicatively. Quadratic filter outputs are squared and then subtracted from other inputs, giving them a suppressive effect on spiking (although quadratic excitation is also possible). **(right)** Cross-validated rate prediction averaged over 167 repeated trials.

## 5.2 Retinal ganglion spike train

The Poisson GLM provides a popular model for neural spike trains due to its ability to incorporate dependencies on spike history (e.g., refractoriness, bursting, and adaptation). These dependencies cannot be captured by models with inhomogeneous Poisson output like the multi-filter LNP model (which is also implicit in information-theoretic methods like MID [21]). The GLM achieves this by incorporating a one-dimensional linear projection of spike history as an input to the model. In general, however, a spike train may exhibit dependencies on more than one linear projection of spike history.

The GQM extends the GLM by allowing multiple stimulus filters and multiple spike-history filters. It can therefore capture multi-dimensional stimulus sensitivity (e.g., as found in complex cells) and produce dynamic spike patterns unachievable by GLMs. We fit a Poisson GQM with a quadratic history filter to data recorded from a retinal ganglion cell driven by a full-field white noise stimulus [28]. For ease of comparison, we fit a Poisson GLM, then added quadratic stimulus and history filters, initialized using a spectral decomposition of the MEL estimate (eq. 16) and then optimized by numerical ascent of the full log-likelihood. Both quadratic filters (which enter with negative sign), have a suppressive effect on spiking (Fig. 5). The quadratic stimulus filter induces strong suppression at a delay of 5 frames, while the quadratic spike history filter induces strong suppression during a $50$ ms window after a spike.

## 6 Conclusion

The GQM provides a flexible class of probabilistic models that generalizes the GLM, the 2nd-order Volterra model, the Wiener model, and the elliptical-LNP model [3]. Unlike the GLM, the GQM allows multiple stimulus and history filters and yet remains tractable for likelihood-based inference. We have derived expected log-likelihood estimators in a general form that reveals a deep connection between likelihood-based and moment-based inference methods. We have shown that GQM performs well on neural data, both for discrete (spiking) and analog (voltage) data. Although we have discussed the GQM in the context of neural systems, but we believe it (and EL-based inference methods) will find applications in other areas such as signal processing and psychophysics.

**Acknowledgments**

We thank the L. Paninski and A. Ramirez for helpful discussions and V. J. Uzzell and E. J. Chichilnisky for retinal data. This work was supported by Sloan Research Fellowship (JP), McKnight Scholar's Award (JP), NSF CAREER Award IIS-1150186 (JP), NIH EY019288 (NP), and Pew Charitable Trust (NP).

## Footnotes

[2]When responses $y_i$ are spike counts, these correspond to the STA and STC.

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
