[Reviews · NeurIPS 2013]

Submitted by Assigned_Reviewer_4

This paper makes two contributions. First, this paper continues a trend of recent work by Park and Pillow (2011) and Ramirez & Paninski (2012) to provide a firm grounding of the spike-triggered methods STA and STC within forward, likelihood-based modelling. STA and STC are calculated as moments from the stimulus/response ensemble, and are popular for constructing stimulus/response models due to their speed and simplicity. However, as explored extensively in previous work (e.g. Paninski 2003, Sharpee et al 2004, Samengo & Gollisch 2013), the validity and utility of these moments depends on particular restrictive conditions on the stimulus set. The prior work of Park & Pillow (2011) was able to shed light on this, by relating these moments to the parameters of a Generalised Linear Model (GLM), a popular forward model for neural responses. While GLM parameter estimation typically requires log-likelihood maximisation over a full dataset (a potentially slow operation), if one instead maximises the expected log-likelihood (EL) under the stimulus distribution, the STA/STC moments provide a fast and asymptotically consistent means to estimate the GLM. Here, the authors extend this work, by asking how moments such as STA and STC can assist in the estimation of a more complex forward model -- the Generalised Quadratic Model (GQM) -- via the EL framework. In doing so, the paper makes its second contribution, in that it introduces the GQM as a new, general class of models for characterising neural responses. The GQM extends the popular Generalised Linear Model (GLM) by allowing for quadratic (rather than just linear) relationships between stimulus and response (together with a point nonlinearity and exponential-family noise).

For Gaussian likelihoods, the authors derive moment-based estimators for the GQM under the conditions of Gaussian-distributed stimuli, and axis-symmetric stimuli. A similar derivation is provided for Poisson likelihoods under a set of assumptions for the stimulus distribution. Low-rank estimates of the parameters are also shown to be asymptotically consistent. Finally, the authors demonstrate an application of these results to real neural data (intracellular voltage and spiking cells), demonstrating a performance gain of the GQM over the GLM. They also show a synthetic example where the vast speed improvements from using moments, as opposed to maximising the likelihood over a full dataset, are evident.

The paper is technically sound, giving thorough derivations of the relationships between the GQM and stimulus/response moments under the set of conditions studied. While this aspect of the work is technical in nature, its significance -- as for that of recent work on this topic raised above -- is in establishing the theoretical groundwork that underlies the workings of a common set of analytical techniques. These results also build incrementally upon these previous papers, although the relationship between the exponentiated-quadratic model presented in Park & Pillow (2011) section 2.2, and that given here in section 3.2, should be made clearer. The GQM may generally be a promising tool for neural characterisation, although this paper does not really provide a thorough enough evaluation of the realistic data requirements to properly assess this.

This paper, in its current form, does fall a little short in its clarity. Part of my confusion in reading it stems from the fact that half of the time the paper is selling the GQM, and half of the time the paper is demonstrating the deeper relationships between moment-based estimators and the GQM parameters. In addition, the title seems to emphasise spectral methods (I assume this refers to the dimensionality reduction in Section 4), but this only figures as a small part of the paper. Because of the jumps in focus between the EL approach in section 3, dimensionality reduction in section 4, and the GQM demo in section 5, it took me a number of reads just to get a handle on the narrative of the paper. Some bracketing commentary at the start and the end of each section to ground the reader would really be helpful. Also, section 3.3 should really be a part of section 4.

Some minor comments:

- line 207: E[x_i . x_j^3] is zero. This should be E[x_i . x_j^2] or E[x_j^3] or something like that in order to be a third moment.

- line 209: what happens when the stimulus is not white? Also, an explanation of how the general fourth-order moments at the end of the expression for Lambda (line 204), reduce to the matrix M = E[x_i^2 . x_j^2] would be helpful (I admit I didn't follow this logic).

- lines 283-292: Typesetting -- mu_x has the subscript in math italics on line 283, but bold in (17) and on line 291.

- Section 4. I'm not sure why the notation here was changed from vector (bold) x to matrix (bold) X, and scalar (italic non-bold) y to matrix (bold) Y. It makes this section needlessly confusing (X and Y are matrices now?) and harder to connect with the work in previous sections. Surely the authors could stick with vector x and scalar y.

- Fig 4: r^2 = 0.55 in the figure legend, but r^2 = 0.50 in the panel.

- line 122: unfinished sentence
Summary: The proposed model, the GQM, is a natural extension of the GLM, and connects very well with moment-based estimators (e.g. the STA/STC) via the EL framework. This is a technically sound paper.

Submitted by Assigned_Reviewer_5

This paper presents an extension of the widely-used GLM characterization of neural firing to a Generalized Quadratic Model (GQM). The authors show how to fit the GQM model using expected log-likelihoods, and apply it to real neural data. The paper is generally clear and well-written.

This is an interesting extension of the GLM model and, as far as I can tell, the fitting methods are solid. In trying to evaluate the impact of this paper, it's unclear to me whether the improvement of GQM over GLM shown in the paper actually matters in practice. In other words, is this an incremental improvement or will the GQM allow us to answer scientific questions that was not possible using the GLM? The first data example (intracellular) uses only a short snippet of data and doesn't compare to the GLM. The second data example (extracellular) shows an improvement in r^2 from 44% to 58%, but it's unclear to me whether this is a small or large improvement. This also seems to be a small dataset (167 trials, and the length of each trial is not stated).

I had some difficulties understanding Section 4:
- 'We propose to use...space basis': It's unclear to me why it makes sense to make this choice
- 'span precisely the same space': I'm having trouble seeing this
- Should the moment-based method used instead of the GQM or as part of the GQM? Should there be some performance comparison between the moment-based method and the GQM on real data?

Minor comments:
- p.3: 'However, the low-rank' is left hanging
- p.5: some of the equation references in the paragraph after equation (14) seem incorrect
Summary: This work extends the widely-used GLM method for neural characterization. The paper is technically solid, but its benefits over existing methods is not entirely clear.

Submitted by Assigned_Reviewer_6

The paper extends previous work on approximate maximum likelihood-based fitting of GLMs to the more general class of Generalized Quadratic Models. While conceptually similar to multi-filter spike-triggered covariance approaches, the GQM nicely inherits spike-history terms from the GLM. The main contribution of this work is in presenting computationally efficient fitting methods based on optimizing the so-called "expected likelihood" function (an approximate version of the true likelihood function). This paper both establishes the GQM as a good model for neural data, and presents a new and efficient fitting procedure.
Previously, [Park and Pillow 2011] and [Ramirez and Paninski 2013] very nicely described and evaluated the expected likelihood framework for estimating GLM parameters. This paper nicely extends their results to GQMs, and this is a valuable contribution, although the performance of the different algorithms/model combinations are poorly explored in the results section.
The paper has parts that are very well written, but feels a bit sloppy/hurried in others. In particular, the Results, figures and the comparisons could be organized better. There are much mixing of methods and results, which would be fine if it were handled a bit better. For instance, Fig 2. is described in Section 3.1, and Fig 3. in Section 3.3. I guess the idea was to put real data in the Results section and simulations in the Methods, but it makes for a tough read as simulations and figures are not as well described.
Minor points:
Line 122 is incomplete. What does the rest of this sentence say? I'm dying to know!
Line 135 mentions but does not define the form of A.
Line 174-180. Perhaps \tilde{a}_ml (and friends) would be better named a_MEL for maximum expected likelihood as in [Ramirez and Paninski 2013]?
Line 255. Where can we see the filter referred to here?
Figure 3 (right). Why do the MELE curves go down? Why does the optimization take *less* time for more samples?
Section 5.1. Why is a "predominantly linear" V1 neuron being fit with a generalized *quadratic* model? I can appreciate that there are still quadratic components, but I can't really tell how important they are. And there does not appear to be a GLM for comparison. Perhaps a V1 complex cell would be a better test? And how well does the GQM fit using exact-ML perform, since the stimulus does not actually come from a gaussian distribution?
Section 5.2. Stimulus history filters suddenly make an appearance here. How are they fit? Surely not by the methods described earlier, which only apply to gaussian or axis-symmetric input distributions?
Summary: The "expected likelihood" based fitting procedure for GLMs is extended to Generalized Quadratic Models for the assumption of axis-aligned stimulus distributions, and gaussian and poisson noise models. An important contribution, however the evaluation is a bit sparse.
Author Feedback

Author rebuttal: We thank the reviewers for their careful reading of our manuscript and many useful suggestions. We first address some points raised by all reviewers, and then consider the concerns of Reviewer 5 and 6 separately.

* Our theoretical contributions:

In our view, there are four primary contributions:

1. We introduce a moment-based method (analogous to STC) for
dimensionality reduction for analog data (e.g., membrane potential
signal)

2. We provide a model-based, unifying framework for dimensionality
reduction for both spiking and analog data

3. We clarify the link between moment-based dimensionality
reduction and ML estimation under GQM models of the data

4. We derive maximum ELL estimators for a broad class of stimulus
distributions

* On significance:

GQM is already becoming popular in the literature: e.g., [McFarland, Cui and Butts 2013] and [Rajan, Marre, and Tkacik 2013] are using the model, not to mention the long-standing popularity of second-order Volterra models in neuroscience. We believe that the details of the GQM model and our spectral estimation procedures (with their vast speed improvements over other techniques) will impact both theoreticians and practitioners.

* Writing and organization issues:

We apologize to all reviewers for deficiencies in the paper's writing and organization. We agree with many of the reviewer's comments and plan to rewrite and reorganize the manuscript accordingly.

In particular, we will:
* Generally improve flow and reinforce the narrative of the text
* Clean up the notation, and make it consistent
* Merge the current Section 3.3 into Section 4
* Place Fig 2 and Fig 3 nearer their discussions in the text, and expand their captions

* Comparison between GLM and GQM for V1 data:

A common concern of the reviewers is that the results section does not provide a systematic comparison between the GQM and GLM. We agree that this comparison should be included in Fig. 4 and its discussion. For the V1 data, GQM systematically provides a better fit than GLM: r^2 GQM:55%, GLM:50% for V1 example shown.
In addition, we note that GQM captures highly skewed distributions evident in actual membrane potential (Vm) recordings, despite symmetric stimulation. GLM only predicts a symmetric Vm distribution from symmetric stimulation. This is particularly important for Vm distributions because the skewed portion of the distribution drives action potential generation, and thus communication to downstream targets, with greatest efficacy.
Similar observations can be made for the case of RGC.

Reviewer 5:

We apologize for ambiguities that may have made it difficult to readSection 4. To clarify:

* 'We propose... space basis': We propose to use the first moment and the eigenvectors of the whitened second moment to estimate a basis of the feature space. This is similar to the technique used in STA/STC analysis, except that in our case Y need not be positive integers. The remainder of the section discusses why the eigenvectors of the whitened covariance span the feature space.

* 'span precisely the same space': Our argument is that the moment-based estimators derived earlier all arise from estimates of the quantity E[YXX^T]. Critically, however, Y was previously assumed to arise from a quadratic nonlinearity, whereas the nonlinearity f in Section 4 is allowed to be arbitrary. We have mistakenly wrote y = f(beta^T x), but it should have been "Y has mean f(beta^T x) and finite variance" which includes the Gaussian and Poisson noise cases. In fact, this argument is more general than necessary for the GQM family. The response-weighted mean and covariance are ("asymptotic") sufficient statistics. This has not to our knowledge been pointed out in the literature. We will clarify this.

* 'Should the moment-based method used instead of the GQM or as part of the GQM?': Moment-based methods are one means by which the GQM parameters may be estimated, via the expected log likelihood (ELL). In fact, we use the moment-based methods to initialize the ML fit for the GQM for Fig 5 (see also response to Reviewer 6).

Reviewer 6:

Reviewer 6 is correct in that the expected log-likelihood trick doesn't apply to the spike-history dependent filters, since we do not control the spike history and do not have an analytic description of the distribution. We would like to clarify that we consider these two aspects to be separate contributions: (1) ELL for a GQM without spike-history (offering novel moment-based formulas that confer a massive speedup over exact ML); (2) incorporating quadratic spike history into a GQM-style model. To the best of our knowledge, both contributions are novel. ELL-GQM (without spike-history) was used only as an initialization for the ML estimate of the GQM model (with spike history) shown in Fig 5. We apologize for not being more clear. In our experience, adding spike history changes the temporal shape of the filters, but leaves the spatial weighting more or less intact, thereby providing a substantial speedup when optimizing the model with spike history.